# Switchable photovoltaic windows enabled by reversible photothermal complex dissociation from methylammonium lead iodide

Lance M. Wheeler [1], David T. Moore [1], Rachelle Ihly [1], Noah J. Stanton[1], Elisa M. Miller [1], Robert C. Tenent[1], Jeffrey L. Blackburn [1] & Nathan R. Neale [1]

Materials with switchable absorption properties have been widely used for smart window applications to reduce energy consumption and enhance occupant comfort in buildings. In this work, we combine the benefits of smart windows with energy conversion by producing a photovoltaic device with a switchable absorber layer that dynamically responds to sunlight. Upon illumination, photothermal heating switches the absorber layer—composed of a metal halide perovskite-methylamine complex—from a transparent state (68% visible transmittance) to an absorbing, photovoltaic colored state (less than 3% visible transmittance) due to dissociation of methylamine. After cooling, the methylamine complex is re-formed, returning the absorber layer to the transparent state in which the device acts as a window to visible light. The thermodynamics of switching and performance of the device are described. This work validates a photovoltaic window technology that circumvents the fundamental tradeoff between efficient solar conversion and high visible light transmittance that limits conventional semitransparent PV window designs.

[1] National Renewable Energy Laboratory, 15013 Denver West Parkway, Golden, CO 80401, USA. Correspondence and requests for materials should be addressed to L.M.W. (email: lance.wheeler@nrel.gov) or to N.R.N. (email: nathan.neale@nrel.gov)

Myriad materials that adopt the oxide and halide forms of the perovskite crystal structure are known to readily accommodate intercalating species to form a rich array of unique compounds[1,2]. The intercalated compounds are stabilized by the formation of ionic[3,4], charge-transfer complex[5,6], van der Waals[7,8] and $\pi$-stacked fluorylaryl-aryl bonds[9]. The weaker of these bonds are reversibly formed and dissociated with small energy input. Lead halide perovskites ($APbX_3$, where A is an organic or alkali metal cation and X is a halide) have demonstrated unprecedented potential as a photovoltaic (PV) absorber[10] and have also shown to reversibly form hydrates[11–14] and other compounds stabilized by charge-transfer complex bonds with nitrogen-[15–17] and oxygen-donor molecules[18–20].

In this work, we leverage the low formation/dissociation energy of the methylammonium lead iodide-methylamine complex ($CH_3NH_3PbI_3 \bullet xCH_3NH_2$)[18] to demonstrate a cohesive switchable PV window that adapts its absorption properties to solar

conditions without pairing separate electrochromic and PV devices[21]. The PV window device utilizes solar photothermal heating to dissociate $CH_3NH_2$ from the $CH_3NH_3PbI_3$ layer, thereby switching from its complexed, bleached (visibly transparent) state to its dissociated, colored (visibly opaque) state. The device is sealed in a closed atmosphere of dilute (2%) $CH_3NH_2$ gas in argon and thus returns to its complexed, bleached state upon removing the solar irradiation and cooling to re-form $CH_3NH_3PbI_3 \bullet xCH_3NH_2$. This phenomenon circumvents the fundamental tradeoff observed in conventional semitransparent PV window designs[22–24], which sacrifice solar-to-electricity power conversion efficiency (PCE) for the high visible light transmittance critical for window performance[25]. Coupled with the cost-effective, scalable solution-phase processing of lead halide perovskites, this technology widely expands the opportunity for energy-efficient PV deployment beyond solar farms and rooftops to glass building facades and vehicles.

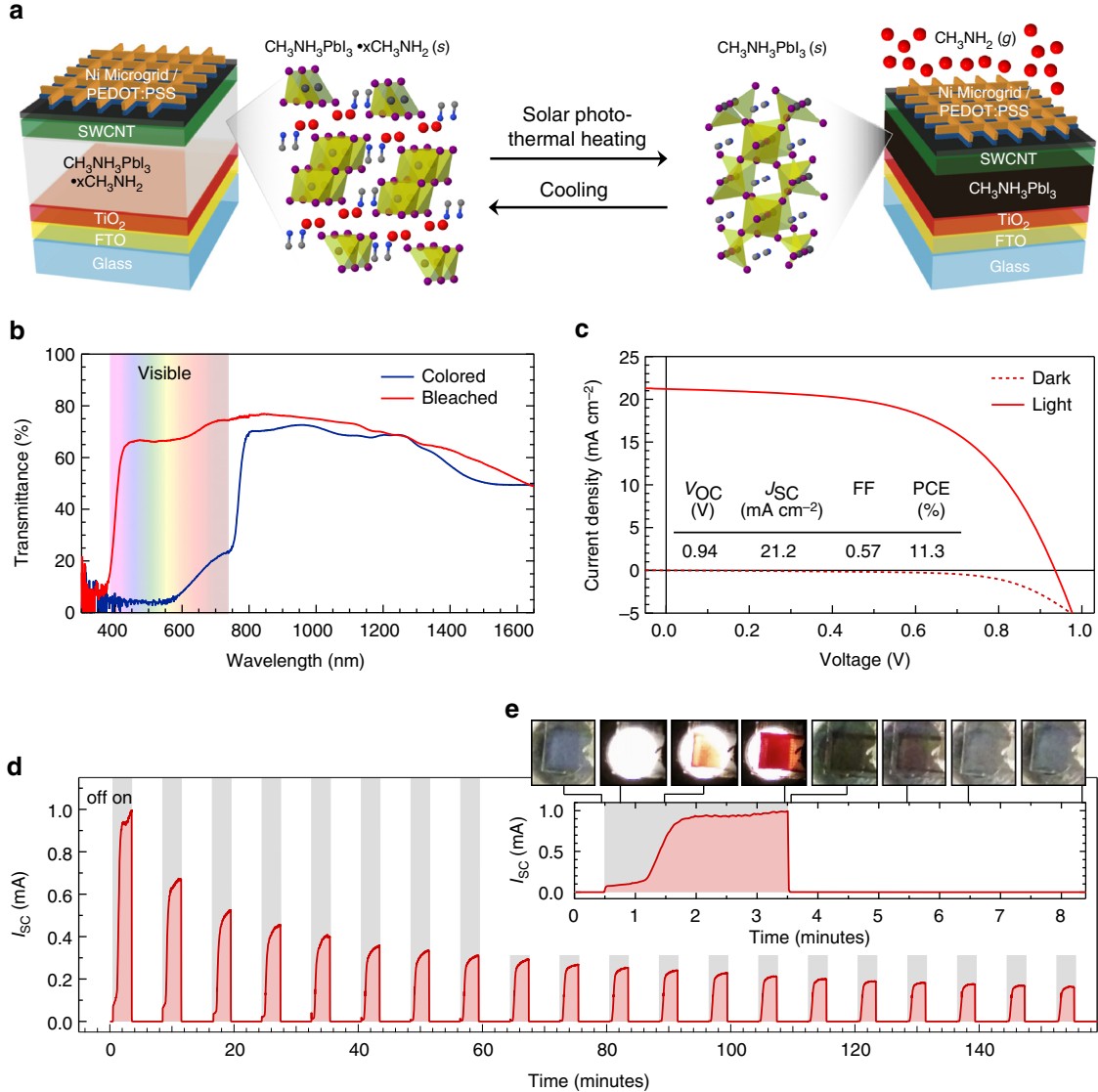

**Fig. 1** Composition and performance of switchable photovoltaic window devices. **a** Schematic of PV window device architecture and switching process. **b** Transmittance of PV devices in the bleached (red) and colored (blue) states as a function of wavelength. **c** Current density as a function of voltage of the champion switchable PV device in the dark (dashed) and under illumination (solid). The inset table shows PV performance metrics of the device before being bleached. **d** Short-circuit current as a function of time for 20 cycles of 3 min of illumination followed by 5 min of cooling in the dark. **e** Short-circuit current as a function of time for the first illumination cycle. The optical images were extracted from Supplementary Movie 1 to show the transition from bleached to colored and back to bleached at the indicated times during the cycling process

## Results

**Switchable PV window design and performance**. The switchable PV window device design and reversible switching mechanism is shown in Fig. 1a, and fabrication details are outlined in the Methods section. Briefly, $CH_3NH_3PbI_3$ was deposited using established methods[26] on titanium dioxide ($TiO_2$) as the electron transport layer and fluorine-doped tin oxide (FTO) as the transparent bottom contact. In order to demonstate switching in a PV device, a number of hole transport and top contact layers were explored. Four complementary layers were required in order to provide the necessary combination of high electrical conductivity, favorable energetic alignment with the $CH_3NH_3PbI_3$ layer, significant transparency in the visible portion of the solar spectrum, and permeability to $CH_3NH_2$ gas (Supplementary Table 1 and Supplementary Fig. 1). We first coated the $CH_3NH_3PbI_3$ with a bilayer of single-walled carbon nanotubes (SWCNTs). SWCNTs wrapped in poly(3-hexylthiophene) (SWCNT/P3HT) provide favorable energetic alignment to the $CH_3NH_3PbI_3$ for hole extraction[27,28], and a second layer of electronically sorted SWCNTs doped with 2,3,5,6-tetrafluoro-7,7,8,8-tetracyanoquinodimethane ($SWCNT^{F4TCNQ}$) was spray-coated to improve lateral electrical transport to the top contact[29]. This layer was then chemically treated with trifluoroacetic acid after deposition to de-polymerize and remove the wrapping polymer to enhance electrical conductivity between the SWCNTs[30]. The device was completed by laminating Ni micromesh[31] coated with poly(3,4-ethylenedioxythiophene):poly(styrenesulfonate) (PEDOT:PSS), an electrically conductive polymer and effective hole transport layer (Supplementary Fig. 2)[32], doped with D-sorbitol that serves as an "electric glue"[33]. Optical and scanning electron microscopy characterization of the device stack is shown in Supplementary Fig. 3.

Figure 1b shows the transmittance of the full device stack spanning the UV to IR portions of the electromagnetic spectrum. The visible portion is highlighted to illustrate reversible switching in this region of the spectrum. The device is highly absorbing in the visible portion in its colored state with an average visible light transmittance of 3%. The visible light transmittance increases to 68% when in the bleached state. The observed decrease in transmittance for both colored and bleached states of the device in the infrared region is due to thin film interference and FTO absorption. Reduced transmittance in this range is desirable for the thermal performance of windows and is the primary function of low-emissivity films used in current high-performance window technology.

Figure 1c shows the current density–voltage curve scanned from high to low positive voltage of the switchable PV device in the dark (dashed) and under 1-sun illumination (solid). The champion device exhibits a PCE of 11.3% with an average of 10.3 ± 0.9% in five devices. A table with the performance metrics of the champion device are inset to Fig. 1c. For comparison, control devices were fabricated with Li-doped 2,2′,7,7′-tetrakis(*N,N*-di-*p*-methoxyphenylamine)-9,9′-spirobifluorene (spiro-OMeTAD) as the hole transport layer and gold as the top contact to mimic conventional $CH_3NH_3PbI_3$-based PV devices[26]. The control devices, which do not exhibit dynamic switching behavior since gold is not permeable to $CH_3NH_2$ gas, exhibit an average PCE of 16.3 ± 0.1%. The short-circuit current density ($J_{SC}$) of 21.2 mA cm$^{-2}$ for the champion switchable device is identical to the control device, whereas the open-circuit voltage ($V_{OC}$) and fill factor (FF) are moderately reduced in the switchable PV window device. Additional characterization of the switchable device (forward and reverse voltage sweeps, stabilized power output, and external quantum efficiency) is included in Supplementary Figs. 4–6.

We next demonstrate dynamic photothermal modulation of the PV window device while generating photocurrent under illumination. Figure 1d is a plot of short-circuit current output as a function of time for a device enclosed in an atmosphere of 2% $CH_3NH_2$ gas balanced with argon to atmospheric pressure. The initial complexed, bleached device was held at ambient temperature from time zero until the device was exposed to solar-simulated illumination at 30 s, indicated by gray boxes in Fig. 1d. Current was immediately observed from the device after illumination, which increased and began to plateau after 1 min. The current dropped to zero when the lamp was turned off after 3 min. The lamp was turned back on after 5 min, and this cycle was repeated 20 times in the closed atmosphere (no additional $CH_3NH_2$ gas was introduced) to demonstrate repeated cycling.

Optical images extracted from Supplementary Movie 1 in Fig. 1e show reversible color changes correlate with current during the cycling process. The exception to this correlation occurred during the initial stages of each illumination cycle, when the device produced current but did not exhibit visible color. After illumination, the current increased at a near-linear rate until 40 s of illumination when the current increased at a higher rate. This sharp rise in current was sustained for another 40 s and was accompanied by a visible color change. After 3 min, the color gradually changed from yellow-orange to dark red, corresponding to complete switching after 3 min. When the lamp was turned off, the device cooled in the chamber, which caused $CH_3NH_2$ gas to intercalate back into the $CH_3NH_3PbI_3$ layer to re-form the $CH_3NH_3PbI_3 \bullet xCH_3NH_2$ complex and return the device to the bleached state after 3 min. The maximum current decreased monotonically from nearly 1 to 0.18 mA after 20 cycles. Supplementary Fig. 7 provides the same analysis as Fig. 1e for the 15th cycle, which exhibits a similar kinetic profile but at a lower current output. Whereas decreased current output may be due to delamination or degradation of transport layers, optical images show inconsistent coloration across the device compared to that following the first cycle, which suggests degradation or morphological changes to the $CH_3NH_3PbI_3$ layer. The source of decreased current after cycling is discussed at length in a following section.

**Thermodynamics of complex formation and dissociation**. We explain the observed reversible complex formation and dissociation process that drives photothermal switching of the PV window device using a simple thermodynamic model. The complexed state is stabilized by weak hydrogen bonds between $CH_3NH_2$ and the organic sublattice of $CH_3NH_3PbI_3$[17], and its conversion to the dissociated state is dependant on the partial pressure ($P$) of $CH_3NH_2$ gas ($g$) and temperature ($T$) of the solid ($s$) phase described by

$$CH_3NH_3PbI_3 \bullet xCH_3NH_2(s) \xleftrightarrow{P,T} CH_3NH_3PbI_3(s) + xCH_3NH_2(g) \quad (1)$$

The methylamine complex $CH_3NH_3PbI_3 \bullet xCH_3NH_2$ has been shown to form a solid at lower $x$ values ($x \approx 1$, 2 based on prior work on hydrates[11]) and a second glassy solvate (liquid) phase at higher values of $x$[17,34,35]. Here we will focus on the solid complex observed at low $CH_3NH_2$ pressures that keeps $x$ low and the complex in the solid regime[17]. As demonstrated in previous work on thermochemical energy storage[36,37], the thermodynamics of Eq. (1) are described by the Clausius–Clapeyron relation where the volume is constant and assumed to be equal to the volume of gas[37,38]:

$$\ln P(T) = \frac{\Delta G_i}{xRT} = -\left(\frac{\Delta H_i}{xR}\right)\left(\frac{1}{T}\right) + \frac{\Delta S_i}{xR} \quad (2)$$

where $R$ is the ideal gas constant (8.31 mol$^{-1}$ K$^{-1}$), $\Delta G_i$, $\Delta H_i$, and

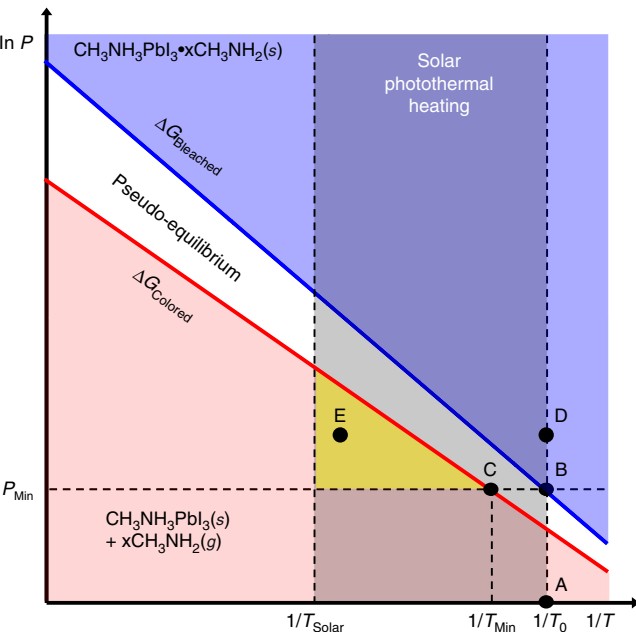

**Fig. 2** Thermodynamic model of complex formation and dissociation. The Clausius-Clapeyron diagram describes the pressure–temperature ($P$-$T$) dependence of $CH_3NH_3PbI_3 \bullet xCH_3NH_2(s)$ formation and dissociation into $CH_3NH_3PbI_3(s) + CH_3NH_2(g)$. Points labeled A—E are described in the text. $P_{Min}$ indicates the minimum pressure needed for $CH_3NH_3PbI_3 \bullet xCH_3NH_2$ formation at room temperature ($T_0$). $T_{Min}$ is the minimum temperature needed for phase transition at $P_{Min}$. The shaded region indicates temperatures attainable by solar photothermal heating up to the maximum temperature, $T_{Solar}$. The yellow region indicates the necessary phase space for achieving switchable PV with solar photothermal heating

$\Delta S_i$ are the Gibb's energy change, enthalpy change, and entropy change, respectively. The subscript $i$ corresponds to the formation of the complexed bleached state or dissociated colored state. Contributions to $\Delta G_i$ thus include complex formation and dissociation, ionic bond formation or destruction in the $CH_3NH_3PbI_3$ lattice, sublimation or condensation, and mixing.

Lines of constant $\Delta G_{Colored}$ and $\Delta G_{Bleached}$ are shown in the Clausius–Clapeyron diagram, which plots $CH_3NH_2$ gas pressure vs. reciprocal temperature (Fig. 2). The lines are separated by a regime of pseudo-equilibrium, which depends on the rates of the forward and reverse reactions[37]. If one of the free parameters (e.g., $P$) is fixed, the variance is zero and phase transition occurs at a certain temperature $T$. The line from point A to point B shows the isothermal transition from the colored state ($CH_3NH_3PbI_3(s)+xCH_3NH_2(g)$) to the bleached state ($CH_3NH_3PbI_3 \bullet xCH_3NH_2(s)$) at room temperature ($T_0$). $P_{Min}$ is the minimum pressure of $CH_3NH_2$ gas needed for full transition to the bleached state. The line from B to C shows an isobaric transition back to the phase-segregated state, which requires a minimum temperature $T_{Min}$. For window applications, the transition temperature must be below what can be practically attained by solar photothermal heating. This is typically less than 75 °C for most climates[5], so this temperature defines $T_{Solar}$ (shaded region in Fig. 2). The switching temperature of conventional vanadium dioxide thermochromic window technology is thus engineered closer to 45 °C for practical application[7].

We investigate the thermodynamics of reversible complex formation illustrated in Fig. 3 using in situ differential Fourier transform infrared spectroscopy (FTIR). A film of $CH_3NH_3PbI_3$ having thickness $W_1$ was deposited onto an attenuated total reflectance (ATR) crystal, and the differential FTIR spectra were

monitored by varying $CH_3NH_2$ gas pressure at ambient temperature as well as modulating temperature at a constant partial pressure (Fig. 3a). Before $CH_3NH_2$ gas was introduced, the film was placed under static vacuum (approximately $40 \times 10^{-3}$ Torr), and the FTIR intensity from this film was zeroed at room temperature to give the red baseline spectrum in Fig. 3b (point A in Fig. 2). $CH_3NH_3PbI_3 \bullet xCH_3NH_2$ formation was not observed by eye or in the FTIR spectrum after the addition of 2 Torr (orange spectrum) and 7 Torr (green spectrum) of $CH_3NH_2$ gas. At 11 Torr, the film bleached and produced positive and negative peaks in the differential FTIR spectrum (point B in Fig. 2, data not shown). The intensity of these peaks remained constant between 11 and 41 Torr $CH_3NH_2$ (blue spectrum, point D in Fig. 2), indicating a $P_{Min}$ of approximately 11 Torr of $CH_3NH_2$ for the room-temperature transition from the colored state to the bleached state. The positive peaks in the 41 Torr spectrum are due to vibrational modes of complexed $CH_3NH_2$. Negative peaks are attributed to loss of intensity from $CH_3NH_3^+$ due to film thickness increase ($W_2 > W_1$) as a result of $CH_3NH_2$ intercalation into the film[39].

Figure 3c shows the isobaric (41 Torr $CH_3NH_2$) evolution of the $CH_3NH_3PbI_3 \bullet xCH_3NH_2$ film as temperature is increased from $T_0$ to 65 °C. We focus on the N–H stretching vibrations of the perovskite A-site cation $CH_3NH_3^+$ and complexed $CH_3NH_2$. Positive intensity from $CH_3NH_2$ is centered at 3250 cm$^{-1}$, and negative intensity due to $CH_3NH_3^+$ is centered at 3130 cm$^{-1}$. The FTIR spectrum is constant from $T_0$ (point D in Fig. 2) up to 35 °C, showing the expected large negative intensity $CH_3NH_3^+$ and positive intensity $CH_3NH_2$ peaks for the $CH_3NH_3PbI_3 \bullet xCH_3NH_2$ film relative to the dissociated $CH_3NH_3PbI_3$ film (baseline). At 40 °C, a threefold reduction in negative $CH_3NH_3^+$ intensity was observed in conjunction with a nearly twofold reduction in positive $CH_3NH_2$ intensity, which both indicate significant dissociation of $CH_3NH_3PbI_3 \bullet xCH_3NH_2$ into $CH_3NH_2(g)$ and $CH_3NH_3PbI_3(s)$ phases. Intensity from all species was further reduced as the temperature was increased to 65 °C (point E in Fig. 2), though the intensity never returned to the baseline level, suggesting that complete dissociation does not occur under these conditions. However, the presence of a clear isosbestic point at 3225 cm$^{-1}$ during this isobaric experiment suggests the lack of intermediates between these two phases. This evidence supports nearly complete and rapid $CH_3NH_2$ dissociation at the threshold temperature, consistent with the Clausius–Clapeyron diagram in Fig. 2 in which the pseudo-equilibrium region is narrow.

Finally, in Fig. 3d we show the isobaric temperature-driven complex formation and dissociation during repeated cycling as demonstrated in the PV window device described in Fig. 1. Two bleached-to-colored cycles were completed by heating the ATR crystal to 60 °C (red spectra) and allowing it to cool to 25 °C (blue spectra) under 11 Torr $CH_3NH_2$. It is clear from the inset optical images that the film visible transmittance switching behavior is correlated to the intensity changes of the N–H stretching vibrations. Importantly, the cumulative observations from these isobaric experiments indicate that for reasonable $CH_3NH_2$ pressures (11–41 Torr), solar thermal temperatures only need to exceed a $T_{Min}$ of ca. 35 °C for switching the PV window into the colored state to generate photocurrent. Such temperatures are readily achievable in most climate zones[5] and can be tuned by the pressure of $CH_3NH_2$ gas to accommodate a desired switching threshold.

**Mechanism of switchable device degradation.** Building-integrated PV must be durable. For instance, existing dynamic smart window technologies are tested out to 50,000 switching cycles using standard durability evaluation methods. Figure 1d

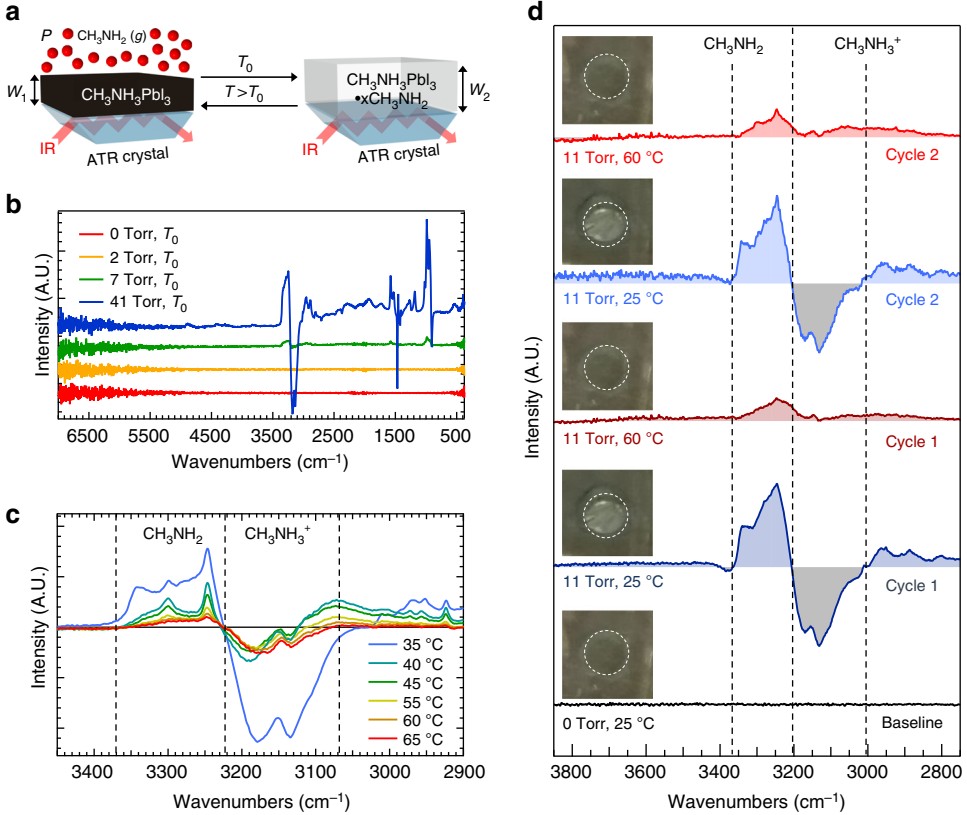

**Fig. 3** In situ differential FTIR of complex formation and dissociation. **a** Diagram illustrating reversible pressure and temperature modulation of complex formation and dissociation on an ATR crystal. Intercalation and complex formation results in an increase in film thickness ($W_2 > W_1$). **b** Differential FTIR spectra at increasing $CH_3NH_2$ pressure. Spectra are offset for clarity. **c** Stretching vibrations due to methylamine ($CH_3NH_2$) and methylammonium ($CH_3NH_3^+$) species for temperatures 35–65 °C at a constant 41 Torr $CH_3NH_2$ pressure. **d** Spectra showing two bleached-to-colored cycles of the $CH_3NH_3PbI_3 \bullet xCH_3NH_2$ film at 20 Torr $CH_3NH_2$. Dashed vertical lines serve to guide the eye. Spectra are offset for clarity. The background due to thin film interference of each spectrum was subtracted with a polynomial fit (Supplementary Fig. 3). Inset images show that the bleached state was observed at 25 °C, and the colored state is achieved at 60 °C. Dashed white circles highlight the ATR crystal. The stage around the ATR crystal is not heated

shows the performance of our demonstration PV device was reduced over the course of 20 switching cycles. We fabricated photoresistors in order to elucidate the degradation mechanism (Fig. 4a). The symmetric FTO lateral contact architecture allows monitoring changes in electrical performance, morphology, and chemistry of the $CH_3NH_3PbI_3$ film without complications arising from the additional gas-porous top contact layers required for a switchable PV device.

Figure 4b shows current as a function of time for a photoresistor composed of $CH_3NH_3PbI_3$ in the absence of $CH_3NH_2$ (0 Torr $CH_3NH_2$) to investigate the effects of photo-thermal cycling without $CH_3NH_3PbI_3 \bullet xCH_3NH_2$ formation and dissociation. The photoresistor was biased at the maximum power point voltage of the PV devices shown in Fig. 1 (0.65 V) and cycled using 3-min photothermal heating followed by 5-min cooling. Current output of the device decreases monotonically by 39% over the course of 20 cycles. X-ray photoelectron spectroscopy (XPS) analysis suggests that this is due to chemical degradation since the film becomes more Pb-rich after cycling (Supplementary Table 2). $CH_3NH_3PbI_3$ is known to evolve $CH_3NH_2$ and hydrogen iodide at elevated temperatures to yield $PbI_2$ within the $CH_3NH_3PbI_3$ film[40], and this degradation pathway is consistent with our observations.

In contrast to $CH_3NH_3PbI_3$, the switchable photoresistor composed of $CH_3NH_3PbI_3 \bullet xCH_3NH_2$ (11 Torr $CH_3NH_2$, Fig. 4c) did not show decreased current output after 20 cycles. The presence of $CH_3NH_2$ is likely beneficial to the stability of the device current output, as $CH_3NH_2$ gas is thought to "heal" defects

in $CH_3NH_3PbI_3$ films[18]. XPS analysis shows the switchable photodector maintains stoichiometry closer to the control sample that was not cycled, suggesting that $CH_3NH_2$ suppresses decomposition to $PbI_2$ via Le Chatelier's principle. The longer switching onset (by ~30 s) and non-saturated current for the $CH_3NH_3PbI_3 \bullet xCH_3NH_2$ (Fig. 4c) relative to the switchable PV device shown in Fig. 1d appears to be due to lower light absorption (~10%, compared to 20–40% for the PV stack), which produces slower photothermal heating for the photoresistor. Longer illumination times led to current saturation and reduced current output that more closely resemble the trend observed in the PV device (Supplementary Fig. 9).

Optical microscopy of the devices after 20 cycles provides the most illustrative insight into the decreased PV device performance. Figure 4d shows an optical microscope image of the 0 Torr $CH_3NH_2$ photoresistor, which is morphologically unchanged from when it was deposited at the scale that can be observed with the microscope. In contrast, the switchable photoresistor (11 Torr $CH_3NH_2$, Fig. 4e) shows roughness on the order of microns to tens of microns. This is an interesting result in comparison to established practice[17,18,35] where $CH_3NH_3PbI_3$ films are briefly exposed to $CH_3NH_2$ gas and annealed in a $CH_3NH_2$-free environment to yield higher quality films. In contrast, $CH_3NH_3PbI_3$ quality is decreased when annealed in presence of a significant partial pressure of $CH_3NH_2$. Morphology is further disrupted with longer illumination times (Supplementary Fig. 9). The loss of current with cycle number provides the most compelling evidence of decreased PV

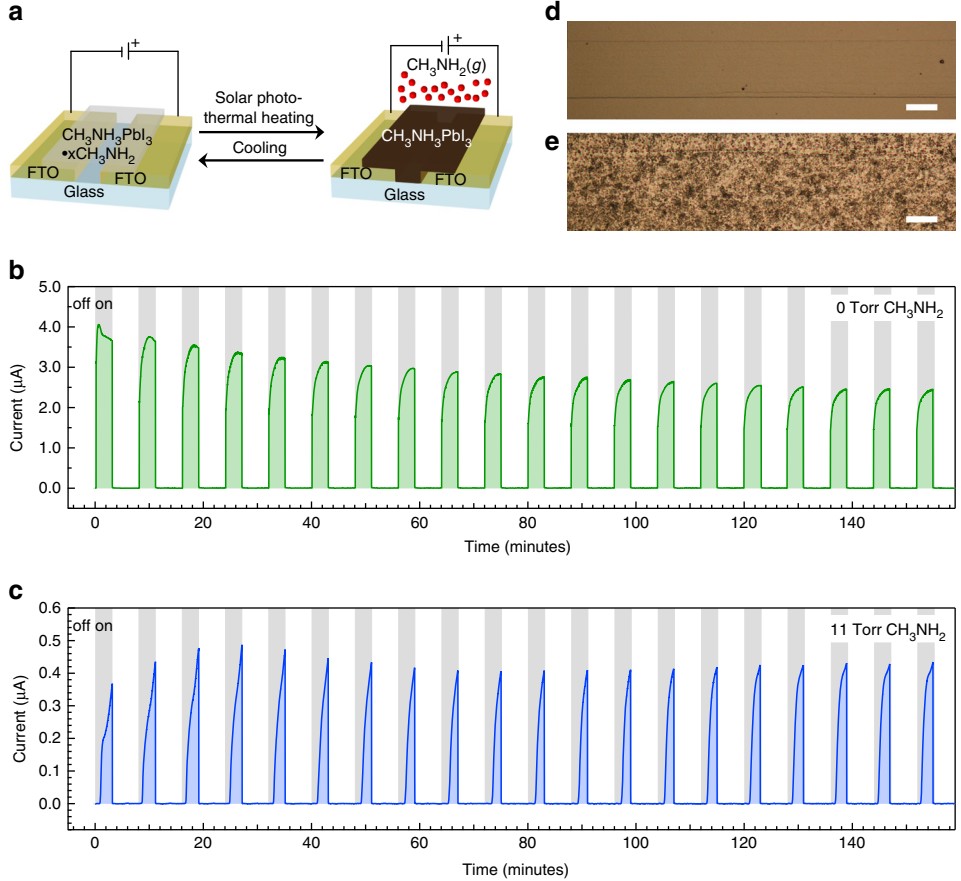

**Fig. 4** Switchable photoresistors to determine PV device degradation. **a** Diagram illustrating switchable photoresistors. **b**, **c** Current as a function of time for 20 cycles of 3-min illumination followed by 5 min of cooling in the dark for a photoresistor in the presence of 0 Torr (**b**) and 11 Torr (**c**) partial pressure of $CH_3NH_2$ gas. Photoresistors were operated at a constant voltage of 0.65 V, which corresponds to the maximum power point voltage of the highest performing switchable PV devices. **d**, **e** Optical photographs of the photoresistor channel after 20 switching cycles in the presence of 0 Torr (**d**) and 11 Torr (**e**) partial pressure of $CH_3NH_2$ gas. Scale bars are 25 μm

performance, whereas the microscope images provide the best window into the mechanism underlying the decrease in performance. Morphology changes likely lead to delamination of adjacent SWCNT and Ni grid hole transport layers, dewetting from the $TiO_2$ layer, and formation of paths for electrical shorting through the device. It is clear that morphological control using device architecture, patterning, or compositional additives, for example, will be necessary in the future development of switchable PV devices based on $CH_3NH_3PbI_3 \bullet xCH_3NH_2$ films.

## Discussion

In this work, we demonstrated a PV window device with optical properties that are photothermally modulated using solar illumination. The devices feature solar energy conversion efficiencies as high as 11.3% in the colored state, high visible light transmittance (68%) in the bleached state, reversible switching over 20 cycles, low switching temperature accessible by solar irradiation in most climates, and fast switching time (less than 3 min). Switching is enabled by reversible dissociation of a methylamine complex with $CH_3NH_3PbI_3$ using solar photothermal energy. The thermodynamics of complex formation and dissociation were described by the Clausius–Clapeyron relation and empirically supported by in situ differential FTIR spectroscopy of the pressure–temperature dependance of the associated phase transitions. Photoresistor studies show the switchable $CH_3NH_3PbI_3 \bullet xCH_3NH_2$ layer has potential for long-term chemical stability, but physical changes to the film during switching

leads to decreased PV device performance over time due to $CH_3NH_2$ loss and disruption of the film morphology. Switchable PV technology circumvents the fundamental tradeoff between power conversion and visible transmittance of conventional semitransparent PV windows and facilitates deployment strategies beyond the current paradigm of rooftops and solar farms.

## Methods

**Absorber layer solution**. The solution used to solubilize $CH_3NH_3PbI_3$ precursors was obtained by charging acetonitrile with $CH_3NH_2$ gas. Acetonitrile was de-gassed using three freeze–pump–thaw cycles and placed in an airtight flask. $CH_3NH_2$ gas was flown into the vessel through a Schlenk-line assembly. The acetonitrile was charged until a 10% by mass $CH_3NH_2$ solution was achieved. The flask was sealed and stored in a −20 °C freezer. Solution was removed with a needle through the septum to keep the solution isolated from air exposure. Adding 1 ml $CH_3NH_2$-charged acetonitrile to 261 mg $PbI_2$ and 87 mg methylammonium iodide formed a typical 0.55 M precursor solution used for devices.

**FTIR measurements**. A Bruker Alpha FTIR spectrometer outfitted with a diamond ATR crystal attachment was used in the study. A $CH_3NH_3PbI_3$ film was deposited onto an ATR crystal by drop-casting a 55 mM solution. Diluting a 0.55 M solution with acetonitrile charged with 10% $CH_3NH_2$ by mass formed the solution. The film was annealed at 100 °C for 30 min in Ar. Securing a custom glass chamber over the ATR crystal stage with a Viton O-ring enclosed the film. The film was pumped down with a roughing pump overnight to obtain a base pressure of 90 mTorr (12 Pa). $CH_3NH_2$ was introduced into the glass jar at varying pressure measured with a Varian type 0531 vacuum gauge. The vacuum gauge was calibrated to measure $CH_3NH_2$ using a 100 Torr range MKS Barotron pressure transducer. In Fig. 3d, the atmosphere was balanced with argon to slightly above atmospheric pressure (627 Torr (83,593 Pa)) measured on a 1000 Torr range MKS Barotron pressure transducer. The temperatures reported are those measured and

delivered to the ATR stage with OPUS 7.2 software. Spectra shown at 60 °C were taken after the 3 min it took to ramp to that temperature. Spectra shown at 25 °C were taken after 21 min, which is the time needed to cool and stabilize at that temperature. The backgrounds of the resulting spectra were nonlinear due to thin film diffraction from change in the $CH_3NH_3PbI_3$ film thickness. The background was subtracted using a third-order polynomial fit using IGOR Pro version 6.37. One fit was used to subtract the baseline at each temperature. The raw data and polynomial fits are shown in Supplementary Fig. 2.

**Photothermal switching of $CH_3NH_3PbI_3 \cdot xCH_3NH_2$(s) films.** Glass was cut into a 1-mm-thick 2.5 cm × 2.5 cm square, sonicated in acetone for 15 min, and blown dry with dry air. The substrates were treated in a UV-ozone cleaner for 15 min before spin-coating a 0.55 M solution composed of methylammonium iodide and $PbI_2$ (5% excess) and the 10% $CH_3NH_2$ in acetonitrile solution onto the substrate at 2000 rpm for 20 s. The film was annealed at 100 °C for 30 min on a hot plate. Photothermal switching of $CH_3NH_3PbI_3$ films was measured in a custom-built glass chamber outfitted with optical ports and feedthroughs for gas input/output and a pressure gauge. The glass substrate with $CH_3NH_3PbI_3$ was cut into 1 cm × 1 cm square. Note: The thermal mass, and thus size, of the substrates plays an important role in switching kinetics. The sample was secured in the chamber with a clip facing the optical port, sealed with a Viton O-ring and clamp, and pumped down overnight to reach a base pressure of 70–90 mTorr (12 Pa), which was measured with a Varian type 0531 vacuum gauge. $CH_3NH_2$ was introduced into the chamber and backfilled with argon to reach slight overpressure above atmospheric pressure (~720 Torr = 95,992 Pa). A Cole-Palmer Illuminator 41,720-series was used for solar-simulated illumination. The lamp was used to approximate solar conditions by adjusting the intensity to between 500 and 1500 W m$^{-2}$ using a Newport power meter (model 841-PE) with a model 818P-015-19 sensor head.

**SWCNT/P3HT ink preparation.** This procedure follows directly from ref.[7]. Powdered SWCNTs produced by the CoMoCAT process, SWeNT CG200, were purchased from Sigma-Aldrich. The producers of this material report a diameter range of 0.7–1.4 nm and a relative purity of 90% as the percentage of carbon that is present as SWNTs. rr-P3HT (3.0 mg; Rieke Metals Inc.; average molecular weight of 50,000 g mol$^{-1}$ and regioregularity of 95%) was dissolved in 5.00 ml of chlorobenzene and sonicated in a bath sonicator for 60 min. SWCNTs (2.5 mg) were added, as purchased, to the dissolved polymer solution and treated with a Cole Parmer 750 W ultrasonic probe, operating at 100% power, for 10 min. After sonication, 5 ml of chlorobenzene was added to improve the solubility of the polymer–nanotube hybrids. The mixture was subsequently centrifuged for 8 min at 10,000 $g$ (Beckman Coulter ultracentrifuge, SW32Ti rotor) to remove unfunctionalized SWNTs and other carbonaceous particles. The precipitate was discarded, and the supernatant was recovered. Ten milliliters toluene was added in order to remove the excess polymer. The mixture was then mildly heated for 60 min to induce aggregation of the functionalized SWNTs. The aggregates were then removed by centrifugation (4 min at 16,000 $g$). The supernatant containing excess polymer was discarded, and the precipitate was recovered. The pellet consisted of 1.5–1.6 mg of polymer-wrapped nanotubes, which were dispersed in 6 ml of chloroform. Immediately prior to spin-coating, the chloroform dispersion was sonicated with an ultrasonic probe for 1 min at low intensity (10% of amplitude) to break up clusters and bundles.

**SWCNT$^{F4TCNQ}$ ink preparation.** SWCNTs were synthesized in-house at NREL via laser vaporization of a graphite target at a furnace temperature of 1125 °C. The imine-based polyfluorene dispersing polymer, poly[(9,9-di-$n$-dodecyl-2,7-fluorendiyl-dimethine)-(1,4-phenylene-dinitrilomethine)] (PFPD), was synthesized in-house as described in ref.[41]. To disperse the SWCNTs, 1.4 mg ml$^{-1}$ SWCNTs and 2 mg ml$^{-1}$ PFPD were added to toluene and processed using an ultrasonic probe for 15 min while the vial was submerged in a bath of dry ice and methanol. Following ultrasonication, the undispersed material is pelleted out via 5 min ultracentrifugation (13,200 rpm, 20°C) using a Beckman Coulter SW32Ti rotor. The supernatant was retained and underwent further ultracentrifugation (20 h, 24,100 rpm, 0 °C) to remove excess PFPD. The resulting pellet, containing highly enriched semiconducting SWCNTs wrapped with PFPD (SWCNT/PFPD), was re-dispersed in neat toluene. The SWCNT/PFPD ink was doped in solution phase by adding 250 µg ml$^{-1}$ using 2,3,5,6-tetrafluoro-7,7,8,8-tetracyanoquinodimethane ($F_4$TCNQ). The bleach of the $S_{11}$ transition in the optical absorption spectrum of the ink after adding $F_4$TCNQ indicates successful doping (Supplementary Fig. 3).

**Switchable PV device fabrication.** Substrates with pre-patterned FTO deposited on glass were purchased from Thin Film Devices, Inc. The glass was 1 mm thick. Thicker glass leads to longer switching kinetics. The substrates were sonicated in acetone for 15 min and blown dry with dry air. The substrates were treated in a UV-ozone cleaner for 15 min before spin-coating 100 µl of a presursor solution prepared by diluting 0.5 ml 75 wt. % titanium diisopropoxide bis(acetylacetonate) in 2-propanol with 6.65 ml 1-butanol at 700 rpm for 10 s, followed by 1000 rpm for 10 s, and finally 2000 rpm for 30 s. The resulting film is placed on a hot plate at 125 °C for at least 2 min to evaporate solvent and then placed in a 500 °C furnace to

sinter into $TiO_2$. 100 µL of the absorber layer solution was spin-coated onto the substrate at 2000 rpm for 20 s in an inert atmosphere glovebox. The film was annealed at 100 °C for 30 min in the glovebox to yield a $CH_3NH_3PbI_3$ layer. The SWCNT/P3HT were deposited onto the $CH_3NH_3PbI_3$ layer by spinning the substrate at 3000 rpm and dropping 300 µl of the SWCNT/P3HT dispersion at a rate of approximately 1 drop every 3 s. SWCNT$^{F4TCNQ}$ thin films were deposited onto the device according to a procedure developed by Tenent et al.[42] using ultrasonic spray deposition. Briefly, $CH_3NH_3PbI_3$ films on $TiO_2$/glass substrates were heated to 130 °C on the stage in the spray chamber. Then, the SWCNT$^{F4TCNQ}$ ink was sprayed using dispersion flow rate of 0.25 ml min$^{-1}$, gas flow rate of 7.0 std l min$^{-1}$, and nozzle power at 0.8 W for 30 coats. After deposition, the films were soaked at 80 °C in a solution of 10 µl ml$^{-1}$ of trifluoroacetic acid (Sigma-Aldrich) in toluene for 30 s, followed by a rinse in neat toluene to fully remove the wrapping polymer, PFPD. Ni micromesh composed of a square network of 14 µm × 14 µm Ni bars with 268 µm × 268 µm holes was purchased from Precision Eforming, Inc. A 2-inch × 4-inch piece of mesh was attached to an aluminum plate using polyimide tape. The aluminum plate is attached to a hot plate with polyimide tape and set to 120 °C. PEDOT:PSS (CLEVIOS PH1000, 1.3% weight aqueous suspension) was purchased from Heraeus. Three milliliters of undiluted PEDOT:PSS was combined with 450 mg D-sorbitol and 136 µl dimethylsulfoxide and stirred with a magnetic stir bar for 15 min. The solution was sprayed onto the micromesh on the hot plate using an airbrush (Master Airbrush Model S68). The PEDOT layer is sprayed with 10 passes at a rate of approximately 1 in. s$^{-1}$ at a distance of 6 in. from the mesh with the airbrush throttle fully open. The PEDOT:PSS$^{D-Sorbitol}$ was annealed for 10 min after spraying and then cooled to room temperature before being cut into ~3 mm × 11 mm strips. Transferring the strips with a tweezers to cover the active area and pressing with gentle finger pressure through a flexible polyethylene terephthalate substrate completes the switchable PV device.

**Photoresistor fabrication.** Substrates with FTO deposited on glass were purchased from Thin Film Devices, Inc. A 100 µm channel was scribed into the FTO using a Q-switched, 532 nm Nd:YAG laser scribing system. The substrates were sonicated in acetone for 15 min and blown dry with dry air. The substrates were treated in a UV-ozone cleaner for 15 min before spin-coating $CH_3NH_3PbI_3$ using the same procedure as the PV devices.

**UV–vis–NIR transmittance measurements.** Measurements were carried out on a Cary-6000i spectrometer with an integrating sphere attachment. Devices were scored and cleaved to roughly 1 cm wide to fit into cuvettes. Silicone oil was placed on the back side of the device to adhere it to the cuvette and avoid additional thin film interference. The cuvette was sealed with a septum, and the atmosphere was removed with a roughing pump through a needle. For the bleached state measurement, 5% $CH_3NH_2$ partial pressure was added to the cuvette and backfilled with argon. The cuvette was filled with argon for the colored state measurement.

**XPS measurements.** XPS data were obtained on a Physical Electronics 5600 photoemission system using monochromatic Al K α (1486.7 eV) radiation.

**PV device measurements.** Devices were tested in a nitrogen-filled glovebox using a Newport Oriel Sol3A solar simulator with a xenon lamp. A calibrated reference solar cell (either GaAs or KG5 filtered Si to minimize the spectral mismatch) was used to set the intensity of the lamp to 1000 W m$^{-2}$ AM1.5 conditions. Current density–voltage scans were taken in both directions with a scan rate of 225 mV s$^{-1}$, step size of 10 mV, delay time of 10 ms, and number of power line cycles of 1. Device area was 0.10 cm$^2$. Devices were measured with and without a metal aperture (0.056 cm$^2$) and produced equivalent current densities. Stabilized power output was measured by holding the device at a constant voltage corresponding to the voltage at the maximum power point of the JV scan. External quantum efficiency (EQE) measurements were taken using a Newport Oriel IQE200.

**Photocurrent measurement in switching devices.** Dynamic photoresponse of PV or photoresistor devices was measured in a custom-built glass chamber outfitted with optical ports, feedthroughs for gas input/output, electrical connection, and a pressure gauge. A PV or photoresistor device was cut into 1 cm × 1 cm squares and secured in the chamber with a clip. The electrical connection to the device was made with alligator clips and fed through the chamber to a Kiethley 2400 sourcemeter interfaced to a computer. The chamber is sealed with a Viton O-ring and clamp and pumped down overnight to reach a base pressure of 70–90 mTorr measured with a Varian type 0531 vacuum gauge. Five percent $CH_3NH_2$ partial pressure is introduced into the chamber and backfilled with argon to reach slight overpressure above atmospheric pressure of ~720 Torr. A Cole-Palmer Illuminator 41,720-series is used for solar-simulated illumination. The lamp was used to approximate solar conditions by adjusting the intensity to between 500 and 1500 W m$^{-2}$ using a Newport power meter (model 841-PE) with a model 818P-015-19 sensor head.

**Crystal structures**. Crystal structures in Fig. 1a were for illustrative purposes only, as the actual structure of $CH_3NH_3PbI_3 \bullet xCH_3NH_2$ was not determined. The images were generated from a .cif file of $CH_3NH_3PbI_3 \bullet H_2O$ from ref. 11 using Jmol open-source software. Hydrogen is not shown.

**Data availability**. All data used in this study are available from the corresponding authors upon reasonable request.

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

## Acknowledgements

The authors would like to thank Bobby To for SEM data, Severin Habisreutinger for SWNT/P3HT preparation, and Talysa Klein for photoresistor substrates. We also thank Jao Van De Lagemaat, Jeff Christians, and Joey Luther for helpful guidance and discussion. D.T.M. acknowledges support from NREL's LDRD Director's Fellowship program. The National Renewable Energy Laboratory's Laboratory Directed Research and Development Program funded R.C.T., D.T.M., and L.M.W. for degradation studies. The U.S. Department of Energy, Office of Science, Office of Basic Energy Sciences, Division of Chemical Sciences, Geosciences, and Biosciences supported all other work under contract number DE-AC36-08GO28308 to NREL.

## Author contributions

L.M.W. designed experiments and thermodynamic model, developed hole transport layers, fabricated and tested PV window devices, performed in situ FTIR experiments, interpreted data, and wrote the manuscript. D.T.M. prepared absorber layer solution, fabricated and tested control PV devices, performed optical microscopy, and contributed to design of photoresistor studies. R.I. developed solution-doped SWCNT ink, spray-coated SWCNT films, and performed subsequent film treatment. N.J.S. prepared the SWCNT ink. J.L.B. contributed to experimental design and top contact development. E. M.M. performed and interpreted XPS experiments. R.C.T. contributed to experimental design and data interpretation. N.R.N. designed experiments, interpreted data, and wrote the manuscript. All authors contributed to editing the manuscript.

## Additional information

**Competing interests:** The authors declare no competing financial interests.

