## [Peer Review File · Nature Communications]

Reviewers' comments:

Reviewer #1 (Remarks to the Author):

In their paper „Switchable Photovoltaic Windows Enabled by Reversible Photothermal Complex Dissociation From Metal Halide Perovskite“, the authors present a window that can be switched between a transparent state and a dark state in which the incident light is absorbed by a perovskite solar cell. The switching can be achieved by changing either the temperature or the pressure. I find the concept innovative, the presentation well written and complete and therefore recommend publication in Nature Communication. However, I have the following questions:

-In Figure 1, the authors present the general concept and demonstrate the switching mechanism. According to the text, switching was achieved by switching the solar simulator on and off. Later in the manuscript, it is explained how either a temperature change or a pressure change can be used to switch the window. Have the authors tried to switch their device in this manner under constant illumination? This would be more realistic for a final window application. Is it possible to achieve a state of intermediate transparency and intermediate efficiency, e.g. by changing the pressure?

-The current of the solar cell shows a fast degradation during the first cycles. Does this decrease level off after the ~3 hours that are shown in Figure 3D, or does it continue?

-What happens if the on/off times are changed, for example if the lamp is running longer than five minutes?

Reviewer #3 (Remarks to the Author):

In this manuscript, Wheeler et al. demonstrate an innovative switchable PV device upon response to the photothermal effect. The key to this new PV device is the low formation/dissociation energy of the methylammonium lead iodide-methylamine complex ($\text{CH}_3\text{NH}_3\text{PbI}_3 \cdot x\text{CH}_3\text{NH}_2$), which was first reported by Zhou et al in the literature (Angew. Chem. Int. Edit. 54, 9705–9709 (2015)). The authors should cite this original reference clearly in the introduction. For example, in the 2nd paragraph of this manuscript, “In this work, we leverage the low formation/dissociation energy of the methylammonium lead iodide-methylamine complex ($\text{CH}_3\text{NH}_3\text{PbI}_3 \cdot x\text{CH}_3\text{NH}_2$) to demonstrate the world’s first switchable PV window that adapts its absorption properties to solar conditions.”

The authors also show the degradation of this new device. Regardless the reversibility of the switchable function, the maximum current that can be achieved is reduced significantly after several cycles. The authors have proposed several possible origins. However, it is important to do more characterization experiments to find out the most possible reason, which will help point out the possible solutions to the significant degradation issue in the future. For example, recrystallization of the liquid-phase $\text{CH}_3\text{NH}_3\text{PbI}_3 \cdot x\text{CH}_3\text{NH}_2$ to MAPbI_3 perovskite causes incomplete film-coverage in the new perovskite film due to high surface energy (or non-wetting) between CNT and the liquid. The authors are encouraged to perform more model experiments to support such assumptions.

Overall, this manuscript includes new findings and promising technological impact, it is suitable for publication in Nature Communications after addressing the above comments.

Response to Reviewer comments:

We thank the reviewers for the overwhelmingly positive reviews. The questions and concerns raised by the reviewers are ones we also had. We have performed experiments to directly address these questions and concerns and believe the manuscript is now a more complete and compelling story for publication. The reviewer comments are included below in *italics*. We directly respond to comments in **red**.

Reviewer #1:

In their paper „Switchable Photovoltaic Windows Enabled by Reversible Photothermal Complex Dissociation From Metal Halide Perovskite“, the authors present a window that can be switched between a transparent state and a dark state in which the incident light is absorbed by a perovskite solar cell. The switching can be achieved by changing either the temperature or the pressure. I find the concept innovate, the presentation well written and complete and therefore recommend publication in Nature Communication. However, I have the following questions:

-In Figure 1, the authors present the general concept and demonstrate the switching mechanism. According to the text, switching was achieved by switching the solar simulator on and off. Later in the manuscript, it is explained how either a temperature change or a pressure change can be used to switch the window. Have the authors tried to switch their device in this manner under constant illumination? This would be more realistic for a final window application.

We agree with the reviewer. Constant illumination is more realistic. We have performed an experiment with one-hour illumination using a switchable photoresistor, and included the current-voltage data and optical image of the film after 3 cycles as Fig. S9. We thank the reviewer for this suggestion. The nicely fits in to our newly-formed understanding of degradation of the switchable PV devices. A discussion was added to the final section of the paper.

Is it possible to achieve a state of intermediate transparency and intermediate efficiency, e.g. by changing the pressure?

Because this is a phase transition described by the Clausius-Clapeyron relation, we do not believe it possible to maintain a state of intermediate transparency. Thinner films will yield more transparent colored state, so the transparency of the colored state can be easily tuned. Changing the pressure will tune the transition temperature, but not the transparency of the final state.

-The current of the solar cell shows a fast degradation during the first cycles. Does this decrease level off after the ~3 hours that are shown in Figure 3D, or does it continue?

In response to this concern and the comments of Reviewer #3, we have included a new section and a fourth figure in the paper dedicated to degradation of the switchable PV

device, entitled “**Mechanism of Switchable Device Degradation**”. Switchable photoresistors were constructed and characterized with electrical performance. The chemical and physical properties of the devices were also characterized with x-ray photoelectron spectroscopy and optical microscopy, respectively. In light of these results, we believe the current will continue at the level achieved after 20 cycles after the morphology of the film has reached a state that is consistently reformed.

-What happens if the on/off times are changed, for example if the lamp is running longer than five minutes?

An experiment with one-hour illumination was performed using a switchable photoresistor. The data is shown in Fig. S9.

Reviewer #3:

In this manuscript, Wheeler et al. demonstrate an innovative switchable PV device upon response to the photothermal effect. The key to this new PV device is the low formation/dissociation energy of the methylammonium lead iodide-methylamine complex ($\text{CH}_3\text{NH}_3\text{PbI}_3 \cdot x\text{CH}_3\text{NH}_2$), which was first reported by Zhou et al in the literature (Angew. Chem. Int. Edit. 54, 9705–9709 (2015)). The authors should cite this original reference clearly in the introduction. For example, in the 2nd paragraph of this manuscript, “In this work, we leverage the low formation/dissociation energy of the methylammonium lead iodide-methylamine complex ($\text{CH}_3\text{NH}_3\text{PbI}_3 \cdot x\text{CH}_3\text{NH}_2$) to demonstrate the world’s first switchable PV window that adapts its absorption properties to solar conditions.”

We agree. This was an unintended oversight. The citation is now in the manuscript where the reviewer suggested.

The authors also show the degradation of this new device. Regardless the reversibility of the switchable function, the maximum current that can be achieved is reduced significantly after several cycles. The authors have proposed several possible origins. However, it is important to do more characterization experiments to find out the most possible reason, which will help point out the possible solutions to the significant degradation issue in the future. For example, recrystallization of the liquid-phase $\text{CH}_3\text{NH}_3\text{PbI}_3 \cdot x\text{CH}_3\text{NH}_2$ to MAPbI_3 perovskite causes incomplete film-coverage in the new perovskite film due to high surface energy (or non-wetting) between CNT and the liquid. The authors are encouraged to perform more model experiments to support such assumptions.

As suggested by the reviewer, we have performed model experiments using photoresistors to narrow down the mechanism of degradation. As mentioned above, there is a new section of the manuscript, entitled “**Mechanism of Switchable Device Degradation**”, dedicated to this topic. The mechanism suggested by the reviewer is actually quite close to what we found.

Overall, this manuscript includes new findings and promising technological impact, it is suitable for publication in Nature Communications after addressing the above comments.

REVIEWERS' COMMENTS:

Reviewer #1 (Remarks to the Author):

The authors have addressed all my comments and questions sufficiently so that I can recommend publication.

Reviewer #3 (Remarks to the Author):

I am satisfied with the authors' revisions, and, thus, I recommend the publication of this manuscript. I have one additional suggestion. Since the "reversible photothermal complex dissociation" phenomenon may not be universal for all metal halide perovskites, it would be more precise to change the "metal halide perovskites" in the title to the "methylammonium lead iodide" that is studied in this work. Related discussion can be seen in a recently published review paper (DOI: 10.1021/acseenergylett.7b00667).

Response to Reviewer comments:

The reviewer comments are included below in *italics*. When needed, we directly respond to comments in **red**.

Reviewer #1:

The authors have addressed all my comments and questions sufficiently so that I can recommend publication.

Reviewer #3:

I am satisfied with the authors' revisions, and, thus, I recommend the publication of this manuscript. I have one additional suggestion. Since the "reversible photothermal complex dissociation" phenomenon may not be universal for all metal halide perovskites, it would be more precise to change the "metal halide perovskites" in the title to the "methylammonium lead iodide" that is studied in this work. Related discussion can be seen in a recently published review paper (DOI: 10.1021/acseenergylett.7b00667).

We agree with the reviewer, and the title has been changed.